# Neutrophil-to-lymphocyte ratio (NLR), platelet-to-lymphocyte ratio (PLR) are more prominent in retinal artery occlusion (RAO) compared to retinal vein occlusion (RVO)

Guanghao Qin[1,2☯], Fang He[3☯], Hongda Zhang[1], Emmanuel Eric Pazo[1], Guangzheng Dai[1], Qingchi Yao[1], Wei He[1], Ling Xu[1]*, Tiezhu Lin[1]*

**1** He Eye Specialist Hospital, Shenyang, Liaoning Province, China, **2** Dalian Medical University, Dalian, Liaoning Province, China, **3** The 8th Medical Center of the PLA General Hospital, Beijing, China

☯ These authors contributed equally to this work.
* 360970814@qq.com (TL); xuling@hsyk.com.cn (LX)

**Data Availability Statement:** All relevant data are within the manuscript and its Supporting information files.

## Abstract

### Aim

To evaluate the association between the value of neutrophil to lymphocyte ratio (NLR), platelet to lymphocyte ratio (PLR), monocyte to high-density lipoprotein ratio (MHR) and the development of retinal artery occlusion (RAO) and retinal vein occlusion (RVO).

### Methods

This retrospective study assessed 41 RAO, 50 RVO and 50 control (age and gender matched senile cataract) participants. The NLR, PLR and MHR parameters of patients' peripheral blood were analyzed. A receiver operating characteristics (ROC) curve analysis and the best cutoff value were used to specify the predictive value of NLR, PLR and MHR in RAO and RVO.

### Results

The NLR, PLR and MHR were significantly higher in RAO group compared to the control group (p<0.001, p<0.001 and p = 0.008; respectively). The NLR, PLR and MHR were also significantly higher in the RVO group compared to the control group (p<0.001, p = 0.001 and p = 0.012, respectively). The NLR and PLR were significantly higher in the RAO group compared to the RVO group (p<0.001 and p = 0.022, respectively). The optimal cut-off value of NLR to predict RAO was >2.99, with 90.2% sensitivity and 100% specificity. The PLR to predict RAO was > 145.52, with 75.6% sensitivity and 80.0% specificity.

### Conclusion

Higher NLR, PLR and MHR are related to the occurrence of RAO and RVO. NLR and PLR are more prominent in RAO compared to RVO.

**Funding:** This study was supported in part by the National Science Foundation of Liaoning Province, China (2020-MS-360), and Science and Technology Program of Shenyang, China (20-205-4-063). The funding organization had no role in the design or conduct of this research.

**Competing interests:** The authors have declared that no competing interests exist.

## Introduction

Retinal vascular occlusion, which comprises retinal artery occlusion (RAO) and retinal vein occlusion (RVO), is the second most frequent kind of retinal vascular disease after diabetic retinopathy [1]. Epidemiological studies have estimated the frequency of RVO in the general population to be between 0.7 and 1.6 percent [2]. While RAO is less common, a recent European study discovered a 7.2/100000 incidence [3]. RAO and RVO have the potential to compromise visual function and perhaps induce permanent and irreversible vision loss in patients. Since various studies have reported a high risk of stroke and other cardiovascular events in patients associated with acute RAO, and the risk of stroke is determined to be highest within the initial few days after the onset of visual loss [4], RAO is considered an ophthalmic emergency and an analogue to ocular stroke. Furthermore, underlying systemic risk factors such as atherosclerosis, diabetes mellitus, hypertension, and ischemic heart disease have been linked to retinal vascular events [1,5]. Although the pathogenic causes of RVO are unknown, it is hypothesized that inflammatory indicators play an important role in the disease's onset. Several inflammatory factors, including IL6, IL8, ICAM-1, MCP-1, IL1-, IL 17-E, and TNF-α, were found to be higher in the aqueous or vitreous of RVO eyes [6–8], prompting the FDA to approve an anti-inflammatory drug (Ozurdex, an intravitreal dexamethasone implant) for treating macular edema following RVO in 2009. As a result, several prospective trials have shown the efficacy and safety of an intravitreal dexamethasone implant for the treatment of RVO-related macular edema [9–11]. Additionally, a ten-year retrospective study of intravitreal dexamethasone implant (Ozurdex) for RVO also discovered that this anti-inflammatory medication had considerable anatomic advantages [12]. The neutrophil to lymphocytes ratio (NLR) has been gaining popularity as a low-cost and easy measure of inflammation in chronic diseases such as acute coronary syndromes, cancer, and Alzheimer's disease [13–17]. Similarly, as a putative biomarker of inflammation, the platelet to lymphocyte ratio (PLR) is linked to the severity of coronary atherosclerosis [18]. Several studies have recently discovered different possible biomarkers for predicting RAO or RVO from normal blood analysis. The connection between NLR and RVO was revealed (Table 1) [19–23]. Furthermore, the monocyte to high-density lipoprotein ratio (MHR), rather than PLR, may be a useful predictor for the development of RAO [24–26].

The goal of this study was to look into the association between various peripheral blood inflammatory markers as NLR, PLR, and MHR and retinal vascular occlusion. We would also like to know if this holds true across RAO and RVO.

## Material and method

### Study design and participants

The Ethics Committee at He Eye Specialist Hospital (Shenyang, China) approved this study based on the Declaration of Helsinki (IRB-2020-K021.01). Patients diagnosed with RAO and RVO between June 2016 and June 2020 at He Eye Specialist Hospital had their medical data reviewed retrospectively.

Fluorescein angiography (FA) (Heidelberg Engineering, Heidelberg, Germany) and high-resolution spectral domain optical coherence tomography (Cirrus HD-OCT 5000, Carl Zeiss, Dublin, California, USA) were used to examine the retina. An experienced retinal specialist (TL) diagnosed RAO and RVO using an indirect ophthalmoscopy exam and/or FA, depending on the location of the retinal vascular obstruction. Patients with central RAO and branch RAO were included in the RAO group, while patients with central RVO and branch RVO were included in the RVO group. To rule out the possibility of clear inflammation, all RAO patients

**Table 1. Different potential biomarkers from routine blood analysis for predicting RAO or RVO reported in literature.**

| Results | Neutrophil levels in RVO patients were higher than in control subjects (5.1±1.9 vs. 3.6±1.0, p<0.001). In RVO patients, lymphocyte counts were lower than in control subjects (2.0±0.7 vs. 2.6±0.9, p = 0.005), the NLR was considerably greater than in control subjects (3.0± 2.7 vs. 1.5 ±0.3, p<0.001). NLR's optimum cutoff value for predicting RVO was >1.89, which has 72.5 percent sensitivity and 100% specificity. | The NLR in the BRVO and control groups was 2.24±0.79 and 1.89±0.64, respectively, with no statistically significant differences between the two groups. | The BRVO group had considerably higher NLR and PLR than the control group (p<0.001). NLR had an AUC of 0.82, and NLR of >2.48 predicted BRVO with a sensitivity of 58% and specificity of 98%. PLR had an AUC of 0.78, and a PLR of >110.2 predicted BRVO with a sensitivity of 72% and specificity of 72%. | RVO patients had reduced lymphocyte counts (p = 0.001) and significantly greater NLR (p = 0.001) and PLR (p = 0.001). The best NLR and PLR cutoff values for predicting retinal vein occlusion were >1.63 and >98.50, respectively. | CRAO patients showed a considerably higher mean NLR (p = 0.009), a cutoff value of >1.62 with the sensitivity and specificity were 83.8 percent and 55.6 percent for NLR was discovered to be a diagnostic tool. | NLR values in patients with RAO were substantially higher than in control subjects (2.85±1.70 vs. 1.63±0.59, p<0.001). NLRs were 3.8 times greater in patients with RAO than in control subjects. | Both NLR and PLR were significantly elevated in RVO | RAO patients had significantly higher values of neutrophils (p = 0.003), RDW (p = 0.0011), NLR (p = 0.0001) and NLR (p = 0.0001). |
|---|---|---|---|---|---|---|---|
| Participants | 40 RVO patients vs. 40 controls | 30 BRVO patients vs. 30 controls | 81 BRVO patients vs. 81 controls | 111 RVO patients vs. 88 controls | 37 CRAO patients vs. 36 controls | 46 RAO patients vs. 51 controls | 1059 RVO patients | 72 RAO patients vs. 72 controls |
| Country | Turkey | Saudi | China | Turkey | Turkey | Turkey | Australia | Italy |
| Authors | Dursun A et al. [20] | Kumral E. et al. [21] | Zhu DD et al. [22] | Şahin M et al. [19] | Soner Guven et al. [24] | Atum M et al. [26] | Liu Z [23] | Pinna A [25] |
| Years | 2015 | 2016 | 2019 | 2020 | 2020 | 2020 | 2021 | 2021 |

RAO = retinal artery occlusion; RVO = retinal vein occlusion; CRAO = central retinal artery occlusion; BRAO = branch retinal artery occlusion; CRVO = central retinal vein occlusion; BRVO = branch retinal vein occlusion; NLR = neutrophil to lymphocytes ratio; PLR = platelet to lymphocyte ratio; MHR = high-density lipoprotein ratio; ROC = receiver operating characteristics; AUC = area under ROC curve; CI = confidence interval.

included in this study were non-arteritic. Fifty cataract patients with normal fundus who were age and gender matched were used as controls.

The presence of classic clinical symptoms was used to diagnose central retinal artery occlusion (CRAO) or branch retinal artery occlusion (BRAO). CRAO diagnostic criteria include: (i) A history of an abrupt loss of eyesight on one side. (ii) Acute retinal ischemia (retinal opacity with cherry red spot or, multiple distributed patches of retinal opacity all over the posterior pole) on initial ocular exam. (iii) The presence of "box-carring" of the blood column in the retinal vessels. (iv) FA performed at initial check-up after sudden onset of loss of vision, shows indications of absence or substantial stasis of the retinal artery circulation. (v) There was no treatment for CRAO [25]. BRAO diagnostic criteria: (i) A history of rapid visual deterioration. (ii) On initial ocular examination, there was evidence of acute retinal ischemia in the distribution of the blocked branch retinal artery. (iii) FA examination soon after onset reveals signs of absence or substantial stasis of circulation in the affected branch retinal artery. (iv) There was no treatment for BRAO [27].

RVO is caused by partial or total occlusion of a retinal vein, and it is characterized according to the location of the blockage. The obstruction of the retinal vein at or posterior to the

optic nerve head is known as central retinal vein occlusion (CRVO), and the total or partial obstruction of a branch retinal vein is known as branch retinal vein occlusion (BRVO) [28]. Criteria for diagnosis: (i) Fundus examination and color fundus pictures demonstrate retinal vein dilatation and tortuosity. (ii) Hemorrhages ranging in severity from the optic nerve head to the retina's outermost periphery. (iii) Hemorrhages that appear as flame-shaped (superficial) or deep blots (ischemic). (iv) There was no treatment for RVO [29].

The criteria for exclusion included infection, giant cell arteritis (C reactive protein (CRP) was used to rule out giant cell arteritis in RAO patients), any blood disease (including anemia, thrombocytopenia and leukopenia), any tumor, autoimmune illness, heart disease, liver and kidney failure, cerebrovascular disease, history of surgery, smoking and drinking, history of trauma, glaucoma, diabetic retinopathy, and other fundus disorders. Patients on any anticoagulant, anti-inflammatory, anti-hyperlipidemia medication, as well as tumor-related treatment or therapy, were also excluded.

## Clinical assessment

All patients underwent a comprehensive ophthalmological examination, including best corrected visual acuity (BCVA), intraocular pressure (IOP), slit-lamp biomicroscopy, and indirect ophthalmoscopy through a dilated pupil. On overnight fasting blood samples from all patients, routine analysis (Auto-blood cell analyzes BC-5180, Mindray, China) and blood lipid and glycemia tests (Auto-chemistry analyzes CS-1200, Mindray, China) were done in the same laboratory. In addition, CRP levels were tested in all RAO patients (CRP-M100, Mindray, China). The NLR, PLR, and MHR were manually calculated. Gender, age, height, weight, type of diabetes, hypertension, height/weight (BMI), and disease onset time were all reported as demographic data.

## Analytical statistics

SPSS statistics software was used for the statistical analysis (ver. 25.0; SPSS Inc., USA). The descriptive statistics utilized were mean ± standard deviation (SD) and percentage (%). The Chi-square test was used to examine categorical data, which was reported as percentages. The Kolmogorov–Simirnov test was used to determine normality. The three groups were tested for homogeneity of variance using ANOVA. The post hoc analysis employed the Bonferroni test and Dunnett T3 based on homogeneity of variance. To determine the sensitivity and specificity of baseline NLR, PLR, and MHR, as well as the optimal cutoff value predicted by RAO and RVO, we used a receiver operating characteristics (ROC) curve analysis. The measurement of the area under the ROC curve predicted validity. The 95 percent confidence interval (CI) was used, and $p < 0.05$ was considered statistically significant.

## Results

The final analysis of the study included 41 RAO (21 females and 20 males, 65.17±12.82 years) and 50 RVO (28 females and 22 males, 63.76±8.83 years) patients. As a control group, 50 senile cataract patients (27 women and 23 men, 65.54±6.70 years) were gathered. When compared to RVO patients, the time it took for illnesses to manifest was much shorter in RAO patients (Table 2).

The parameters of the blood test were presented in three groups in Table 3. The NLR, PLR, and MHR in the RAO group were significantly higher than in the control group ($p<0.001$, $p<0.001$, and $p = 0.008$; respectively). The RVO group had significantly higher NLR, PLR, and MHR than the control group ($p<0.001$, $p = 0.001$, and $p = 0.012$; respectively). The NLR and

**Table 2. Baseline characteristics of participants.**

|  | RAO (n = 41) | RVO (n = 50) | Control (n = 50) | P value |
|---|---|---|---|---|
| Age (year) | 65.17±12.82 | 63.76±8.83 | 65.54±6.98 | 0.626 |
| Gender (male-%) | 20 (48.8) | 22 (44.0) | 23 (46.0) | 0.901 |
| Hypertension (n- %) | 13 (31.7) | 14(28.0) | 13(26.0) | 0.833 |
| Diabetes (n- %) | 7(17.1) | 8(16.0) | 7(14.0) | 0.918 |
| BMI (kg/m$^2$) | 23.95±4.44 | 24.76±2.63 | 24.10±3.27 | 0.478 |
| Onset time (day) | 7.34±11.97 | 57.96±59.95 |  | <0.001 |

RAO: Retinal artery occlusion, RVO: Retinal vein occlusion, BMI: Height/weight.

PLR in the RAO group were substantially greater than in the RVO group (p<0.001 and p = 0.022, respectively).

The area under the curve of NLR, PLR, and MHR in RAO patients was 0.980, 0.837, 0.694, (Fig 1), while the area under the curve of NLR, PLR, and MHR in RVO patients was 0.739, 0.688, 0.685, according to the ROC curve analysis. (Fig 2) With 90.2 percent sensitivity and 100.0 percent specificity, the best NLR cut-off value for predicting RAO was >2.99. The PLR for predicting RAO was > 145.52, with a sensitivity of 75.6 percent and a specificity of 80.0 percent. The MHR for predicting RAO was > 0.20, with a sensitivity of 80.5 percent and a specificity of 56.0 percent. The NLR for predicting RVO was >1.75, with a sensitivity of 86.0 percent and a specificity of 56.0 percent. The PLR for predicting RVO was > 132.43, with a sensitivity of 66.0 percent and a specificity of 72.0 percent. The MHR for predicting RVO was > 0.20, with a sensitivity of 74.0 percent and a specificity of 56.0 percent.

**Table 3. The comparison of parameters of blood test among three groups.**

|  | RAO (n = 41) | | RVO (n = 50) | | Control (n = 50) | |
|---|---|---|---|---|---|---|
|  | Mean± SD | p$^+$value | Mean ± SD | p$^{++}$value | Mean ± SD | p$^*$value |
| White blood cell count (10$^9$/μl) | 8.71±2.57 | <0.001 | 6.10±1.63 | 0.593 | 5.75±1.47 | <0.001 |
| Neutrophil count (10$^9$/μl) | 6.94±2.28 | <0.001 | 3.98±1.32 | 0.023 | 3.33±1.05 | <0.001 |
| lymphocyte count(10$^9$/μl) | 1.30±0.40 | <0.001 | 1.64±0.56 | 0.008 | 1.95±0.50 | 0.005 |
| Platelet count (10$^9$/μl) | 228.24±56.93 | 1.00 | 230.68±47.83 | 1.00 | 226.34±47.46 | 1.00 |
| Monocyte count (10$^9$/μl) | 0.44±0.20 | 0.003 | 0.40±0.13 | 0.004 | 0.32±0.11 | 0.538 |
| HDL (mg/dl) | 1.62±0.25 | 0.935 | 1.58±0.40 | 0.685 | 1.65±0.31 | 0.914 |
| RDW (%) | 12.71±0.58 | 1.00 | 12.44±0.58 | 0.282 | 12.64±0.65 | 0.092 |
| MPV (fL) | 7.38±2.00 | 1.00 | 7.17±1.54 | 1.00 | 7.19±1.80 | 0.072 |
| LDL(mg/dl) | 3.65±0.95 | 1.00 | 3.56±0.96 | 1.00 | 3.67±1.09 | 1.00 |
| Total cholesterol (mg/dl) | 5.65±1.39 | 1.00 | 5.25±1.04 | 0.995 | 5.48±1.09 | 0.329 |
| Triglyceride (mg/dl) | 1.62±0.71 | 0.651 | 1.86±0.90 | 1.00 | 1.86±1.10 | 0.665 |
| Fasting glucose (mg/dl) | 6.50±2.28 | 0.984 | 6.00±1.49 | 0.529 | 6.36±1.47 | 0.550 |
| NLR | 5.63±2.07 | <0.001 | 2.64±1.20 | <0.001 | 1.77±0.56 | <0.001 |
| PLR | 188.81±64.36 | <0.001 | 154.10±53.83 | 0.002 | 121.65±33.56 | 0.022 |
| MHR | 0.28±0.12 | 0.008 | 0.27±0.11 | 0.012 | 0.21±0.10 | 1.00 |

RAO: Retinal artery occlusion, RVO: Retinal vein occlusion, SD: Stand deviation RDW: Red cell distribution width, MPV: Mean Platelet Volume, HDL: High-density lipoprotein, LDL: Low-density lipoprotein, NLR: Neutrophil to lymphocyte ratio, PLR: Platelet to lymphocyte ratio, MHR: Monocyte to high-density lipoprotein ratio, p$^+$: RAO compared to Controls, p$^{++}$: RVO compared to Controls, p$^*$: RAO compared to RVO.

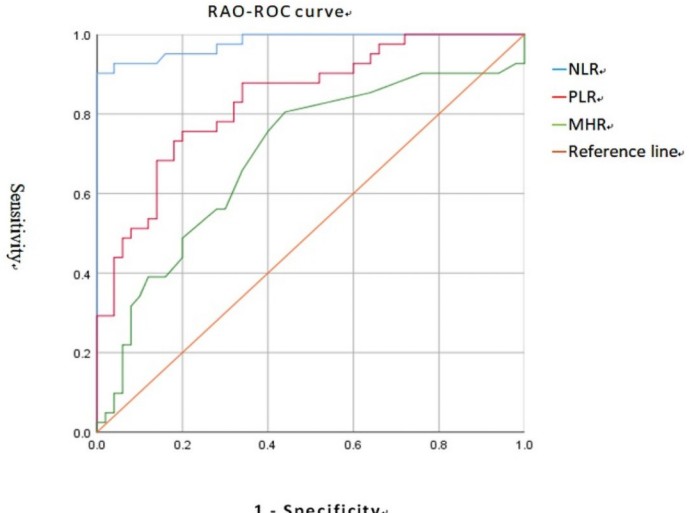

**Fig 1. ROC analysis of NLR, PLR and MHR for RAO.**

## Discussion

The presence of inflammation, tumors, or cardiovascular disease can be indicated by an increase in white blood cells and their many subtypes in the body [30,31]. Neutrophils have been shown to play a major part in the inflammatory response to acute stress, which can cause damage to many tissues and organs in the body. Recent research suggests that increased neutrophil counts contribute to tumor growth and metastasis by releasing chemokines (CK) like IL-8, angiopoietin-1, matrix metalloproteinase-9, and vascular endothelial growth factor (VEGF), which are responsible for the formation of macular edema in some retinal vascular diseases [32,33]. In this study, we discovered that neutrophil levels in the RAO group were considerably greater than those in the control group, which is consistent with previous studies

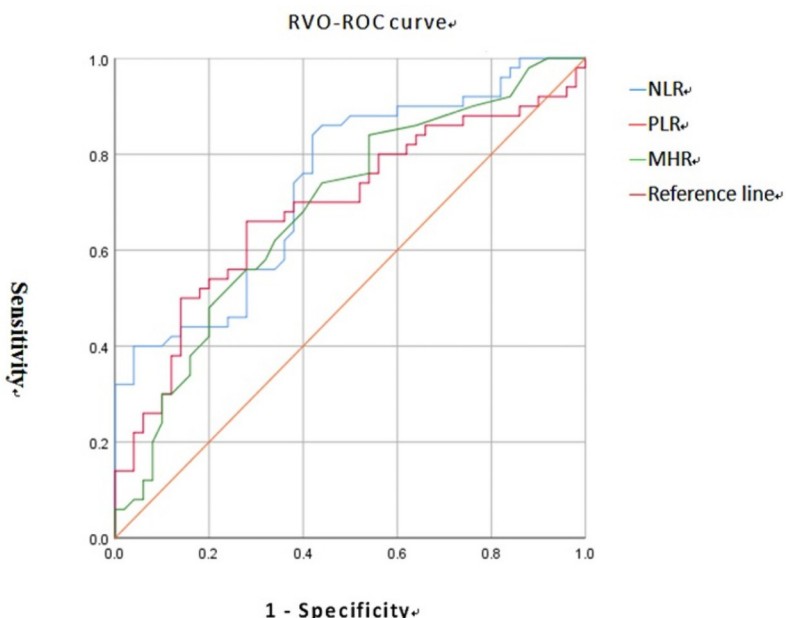

**Fig 2. ROC analysis of NLR, PLR and MHR for RVO.**

[24–26]. Additionally, we discovered that neutrophil levels were significantly higher in the RVO group than in the control group in this study. Although that outcome is consistent with the findings of Zhu et al [22], the other studies have found that the neutrophil values in the RVO group were not significantly different from those in the control group [19,34]. We hypothesized that this might be due to sampling error. Furthermore, numerous systemic examination results should be considered, as they may have an effect on the patients' blood tests. We also discovered that neutrophil counts in the RAO group were considerably greater than those in the RVO group, which has never been reported before. Though inflammation played a role in both RAO and RVO, RAO necessitates immediate medical attention because the disease is usually in its acute stage when patients visit the doctor. In this study, the mean interval time between visits to the doctor was significantly shorter in the RAO group compared to the RVO group. Neutrophophil extracellular traps (NETs) have been discovered to promote thrombus formation by acting as a scaffold for platelets and coagulation activation in several investigations [35]. Although certain histological studies showed NETs in various types of arterial and venous thrombi [36–41], the relative involvement of NETs in thrombosis is still unknown. Mangold et al. [39] proposed that arterial thrombi included more NETs than venous thrombi after analyzing NETs of 30 coronary artery thrombi and 7 deep vein thrombi. Because there is such a big difference in patients between the two groups, a larger study with a greater number of venous thrombi is required to fully appreciate the role of NETs.

Lymphocytes, on the other hand, play the opposite role to neutrophils, regulating the inflammatory response and being a powerful predictor of cardiovascular risk [42]. According to Sonmez et al. [13], lymphocyte counts can be used as a measure for general bad health and physiological stress. In the present study, we discovered that lymphocyte levels in the RAO and RVO groups were significantly lower than those in the control group, which was consistent with earlier research [19,22,24,26,43]. In this study, lymphocyte levels in the RAO group were likewise considerably lower than those in the RVO group. We believe that the acute RAO stage is to blame.

The RAO and RVO groups had considerably greater monocyte counts than the control group in this study. Monocytes are hypothesized to contribute to the formation of atherosclerotic plaques within the vasculature by being implicated in a malignant cascade of endothelial dysfunction, oxidative stress, and inflammation [44–46]. In several research, MHR has been suggested to be a novel inflammatory marker [47,48]. MHR was considerably greater in the RAO and RVO groups compared to the control group in the current study, indicating that MHR could be a biomarker of inflammation in retinal vascular occlusion. However, it was difficult to discriminate MHR values between the RAO and RVO groups. Although some previous studies reported that RDW and MPV might be the biomarkers of RVO and RAO [34,43], we did not find any significant difference among RAO, RVO and control groups. We speculated that the patients' race, exclusion criterion and sample size might be responsible for the various results of different studies.

Prior researches have looked at the effects of NLR and PLR on predicting RAO or RVO [19–26,43,49,50]. In some investigations, NLR has already been shown to be a unique marker for predicting RAO [24,26]. Furthermore, PLR was also predictive of RAO in this investigation, which contradicts prior findings. We also emphasized that NLR had the best ROC, sensitivity, and specificity in predicting RAO. Most studies found that NLR and PLR had predictive value for RVO [19,22], which matched the findings of this investigation. However, the accuracy of NLR and PLR in predicting RVO was lower than that of RAO, which was based on ROC, sensitivity, and specificity.

The study's shortcomings include the fact that it is a retrospective, non-randomized design study with only Chinese participants. Second, due to the small number of patients included,

we were unable to conduct analysis on the subgroup with central or branch occlusion. Third, we did not look at the relationship between various leukocytes and long-term retinal vascular occlusion healing. To sum up, increases in other inflammatory markers such as C-reactive protein, IL-6, and MCP-1 were not examined in this study. Furthermore, various systemic examination results should be considered because they may have an impact on the patient's performance. This is the only study that we are aware of that compares the predictive value of NLR, PLR, and MHR in RAO and RVO.

Although the pathophysiology of RAO and RVO are not the same, previous studies believed that they were both related to thrombosis and inflammation. Some studies suggest that arterial thrombosis contains more NETs than venous thrombosis [39]. Although previous studies have shown NLR and/or PLR might be the potential biomarkers in RAO or RVO [19–26], we would like to explore if there is some difference between them. Finally, we found that in RAO, NLR and PLR were more significant than in RVO. Because of the varying outcomes of PLR, NLR may be a more reliable biomarker for RAO than PLR. To better understand the link between retinal vascular disease and blood biomarkers, larger sample sizes and multi-center research are needed.

## Supporting information

**S1 Dataset. Anonymised data of all patients.**
(XLSX)

## Author Contributions

**Conceptualization:** Ling Xu, Tiezhu Lin.

**Data curation:** Guanghao Qin, Hongda Zhang, Qingchi Yao.

**Formal analysis:** Guanghao Qin, Hongda Zhang, Emmanuel Eric Pazo.

**Funding acquisition:** Ling Xu, Tiezhu Lin.

**Investigation:** Guanghao Qin, Fang He, Guangzheng Dai, Qingchi Yao, Wei He, Tiezhu Lin.

**Methodology:** Fang He, Emmanuel Eric Pazo, Guangzheng Dai, Qingchi Yao, Tiezhu Lin.

**Project administration:** Tiezhu Lin.

**Resources:** Wei He, Ling Xu, Tiezhu Lin.

**Software:** Guanghao Qin, Guangzheng Dai.

**Supervision:** Wei He, Ling Xu, Tiezhu Lin.

**Validation:** Wei He, Ling Xu.

**Visualization:** Emmanuel Eric Pazo.

**Writing – original draft:** Fang He.

**Writing – review & editing:** Emmanuel Eric Pazo, Tiezhu Lin.

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
