## [Decision Letter · Decision Letter 0]

14 Dec 2021

PONE-D-21-18118Neutrophil-to-lymphocyte Ratio (NLR), Platelet-to-lymphocyte Ratio (PLR) are more prominent in retinal artery occlusion (RAO) compared to retinal vein occlusion (RVO).PLOS ONE

Dear Dr. Lin,

Thank you for submitting your manuscript to PLOS ONE. After careful consideration, we feel that it has merit but does not fully meet PLOS ONE’s publication criteria as it currently stands. Therefore, we invite you to submit a revised version of the manuscript that addresses the points raised during the review process.

The manuscript has been evaluated by two reviewers, and their comments are available below. The reviewers have raised a number of concerns that need attention. They request additional information on methodological aspects of the study and the interpretation of the results. Could you please revise the manuscript to carefully address the concerns raised?

We look forward to receiving your revised manuscript.

Kind regards,

Dario Ummarino, Ph.D.

Senior Editor

PLOS ONE

Journal Requirements:

Reviewers' comments:

Reviewer's Responses to Questions

**Comments to the Author**

1. Is the manuscript technically sound, and do the data support the conclusions?

Reviewer #1: Yes

Reviewer #2: Yes

2. Has the statistical analysis been performed appropriately and rigorously? 

Reviewer #1: Yes

Reviewer #2: Yes

3. Have the authors made all data underlying the findings in their manuscript fully available?

Reviewer #1: Yes

Reviewer #2: No

4. Is the manuscript presented in an intelligible fashion and written in standard English?

Reviewer #1: Yes

Reviewer #2: Yes

5. Review Comments to the Author

Reviewer #1: Comments to the Author,

In the present study, Lin T et al. evaluated the association between the value of neutrophil to lymphocyte ratio (NLR),

platelet to lymphocyte ratio (PLR), monocyte to high-density lipoprotein ratio (MHR) and the development of retinal artery occlusion (RAO) and retinal vein occlusion (RVO).

Generally paper is well-written and provides clear explanations for their results. However, I have some important suggestions must be addressed by authors, which contribute to increase the validity of this study as follow;

1) RDW is also one of the important inflammation biomarker. It would be better if authors mention the RDW levels in table 3.

2)Did the authors exclude the patients with anemia?

3) Please add and discuss below references in the discussion part since they are all about your article issue.

1: Kurtul BE, Ozer PA. Neutrophil-to-lymphocyte ratio in ocular diseases: a systematic review. Int J Ophthalmol. 2019 Dec 18;12(12):1951-1958. doi:10.18240/ijo.2019.12.18. PMID: 31850181; PMCID: PMC6901879.

2: Kurtul BE, Çakmak Aİ, Elbeyli A, Özarslan Özcan D, Özcan SC, Cankurtaran V.Assessment of platelet-to-lymphocyte ratio in patients with retinal vein occlusion. Ther Adv Ophthalmol. 2020 Nov 18;12:2515841420971949. doi: 10.1177/2515841420971949. PMID: 33283155; PMCID: PMC7686591.

3: Elbeyli A, Kurtul BE, Ozcan DO, Ozcan SC, Dogan E. Assessment of Red Cell Distribution Width, Platelet/lymphocyte Ratio, Systemic Immune-inflammation Index, and Neutrophil/lymphocyte Ratio Values in Patients with Central Retinal Artery Occlusion. Ocul Immunol Inflamm. 2021 Sep 15:1-5. doi: 10.1080/09273948.2021.1976219. Epub ahead of print. PMID: 34524949.

4) The study would benefit from minor English editing.

Reviewer #2: In this paper, Qi et al. compare NLR, PLR and MHR in RAO and RVO. The Authors' findings are interesting; however, there are several issues which need to be addressed, as outlined below.

Table 1

- Two recent studies, one assessing NLR and PLR role in RVO (Liu et al. Acta Ophthalmol. 2021 Jul 4. doi: 10.1111/aos.14955) and the other investigating CBC measures in RAO (Pinna et al. Acta Ophthalmol. 2021 Sep;99(6):637-643. doi: 10.1111/aos.14699) are missing. These should be included and commented.

- The Table should be displayed in a horizontal view. This makes it easier to read.

Methods

C-reactive protein and/or erythrocyte sedimentation rate (ESR) are essential to make a diagnosis of arteritic RAO. However, these markers were not analyzed in this paper (line 207). The Authors state that giant cell arteritis was a criterion for exclusion (line 110). How was this exclusion made? If there are no patients with arteritic RAO, all the subjects enrolled in the study had non-arteritic RAO. If so, this has to be clearly stated throughout the manuscript.

Indeed, if in the RAO group there are patients with both the arteritic and the non-arteritic form, the higher NLR values associated with arteritic RAO may significantly affect the final overall result. This is a critical point to clarify.

Discussion

- RAO and RVO are two different vascular retinal disorders with different pathophysiology, risk factors and natural history. What's the rationale for comparing NLR, PLR, and MHR values in these vasculopathies? Please explain.

- line 171. Some recent studies suggest lack of association between NLR and PLR and RVO. Conversely, MPV and, to a lesser extent, RDW may be disease biomarkers in RVO (Pinna et al. Ophthalmic Epidemiol. 2021 Feb;28(1):39-47. doi: 10.1080/09286586.2020.1791349). This should be discussed.

6. PLOS authors have the option to publish the peer review history of their article (what does this mean?). If published, this will include your full peer review and any attached files.

Reviewer #1: No

Reviewer #2: No

---

## [Author Response · Author response to Decision Letter 0]

10 Jan 2022

RE: PONE-D-21-18118

Title: Neutrophil-to-lymphocyte Ratio (NLR), Platelet-to-lymphocyte Ratio (PLR) are more prominent in retinal artery occlusion (RAO) compared to retinal vein occlusion (RVO).

Reviewer #1 comments: 

1) RDW is also one of the important inflammation biomarkers. It would be better if authors mention the RDW levels in table 3.

Response: We thank the reviewer for the comments. We have added the relevant data about RW and MPV in Table 3, but we didn’t find any significant difference among 3 groups. (page 12, lines 197-200).

2)Did the authors exclude the patients with anemia?

Response: We thank the reviewer for the comments. In this study we excluded the patients with any blood disease, including anemia, thrombocytopenia and leukopenia. (pages 6, lines 110).

3) Please add and discuss below references in the discussion part since they are all about your article issue.

1: Kurtul BE, Ozer PA. Neutrophil-to-lymphocyte ratio in ocular diseases: a systematic review. Int J Ophthalmol. 2019 Dec18;12(12):1951-1958. doi:10.18240/ijo.2019.12.18. PMID: 31850181; PMCID: PMC6901879.

2: Kurtul BE, Çakmak Aİ, Elbeyli A, Özarslan Özcan D, Özcan SC, Cankurtaran V.Assessment of platelet-to-lymphocyte ratio in patients with retinal vein occlusion. Ther Adv Ophthalmol. 2020 Nov 18;12:2515841420971949. doi:10.1177/2515841420971949. PMID: 33283155; PMCID: PMC7686591.

3: Elbeyli A, Kurtul BE, Ozcan DO, Ozcan SC, Dogan E. Assessment of Red Cell Distribution Width, Platelet/lymphocyte Ratio,Systemic Immune-inflammation Index, and Neutrophil/lymphocyte Ratio Values in Patients with Central Retinal Artery Occlusion. Ocul Immunol Inflamm. 2021 Sep 15:1-5. doi: 10.1080/09273948.2021.1976219. Epub ahead of print. PMID:34524949.

Response: We thank the reviewer for the comments. We have added the reference above in the paper. (references 43,49,50).

4) The study would benefit from minor English editing.

Response: We thank the reviewer for the comment. We have done the English editing.

Reviewer #2 comments:

Table 1

- Two recent studies, one assessing NLR and PLR role in RVO (Liu et al. Acta Ophthalmol. 2021 Jul 4. doi: 10.1111/aos.14955) and the other investigating CBC measures in RAO (Pinna et al. Acta Ophthalmol. 2021 Sep;99(6):637-643. doi:10.1111/aos.14699) are missing. These should be included and commented.

- The Table should be displayed in a horizontal view. This makes it easier to read.

Response: We appreciate the reviewer's comments. Table 1 has been updated to include the two studies. Additionally, we reoriented Table 1 horizontally.

Methods

C-reactive protein and/or erythrocyte sedimentation rate (ESR) are essential to make a diagnosis of arteritic RAO. However, these markers were not analyzed in this paper (line 207). The Authors state that giant cell arteritis was a criterion for exclusion (line 110). How was this exclusion made? If there are no patients with arteritic RAO, all the subjects enrolled in the

study had non-arteritic RAO. If so, this has to be clearly stated throughout the manuscript.

Indeed, if in the RAO group there are patients with both the arteritic and the non-arteritic form, the higher NLR values associated with arteritic RAO may significantly affect the final overall result. This is a critical point to clarify.

Response: We thank the reviewer for this great advice. In this study, we have excluded the patients with giant cell arteritis. CRP was tested in only RAO patients. (page 5-6, lines 89-90，109-110). 

Discussion

- RAO and RVO are two different vascular retinal disorders with different pathophysiology, risk factors and natural history. What's the rationale for comparing NLR, PLR, and MHR values in these vasculopathies? Please explain.

- line 171. Some recent studies suggest lack of association between NLR and PLR and RVO. Conversely, MPV and, to a lesser extent, RDW may be disease biomarkers in RVO (Pinna et al. Ophthalmic Epidemiol. 2021 Feb;28(1):39-47. doi:10.1080/09286586.2020.1791349). This should be discussed.

Response: We thank the reviewer for the comments. Although the pathophysiology of RAO and RVO are not the same, previous studies believed that they were both related to thrombosis and inflammation. Some studies suggest that arterial thrombosis contains more NETs than venous thrombosis. (pages 11, lines 175-184) Although previous studies have shown NLR and/or PLR might be the potential biomarkers in RAO or RVO, we would like to explore if there is some difference between them. We also included the relevant data for RW and MPV in Table 3, but we didn’t find any significant difference among the 3 groups. We speculated that the patients’ race, exclusion criterion and sample size might be responsible for the various results of different studies. (pages 12, lines 197-200).

---

## [Editor Report · Decision Letter 1]

14 Jan 2022

PONE-D-21-18118R1Neutrophil-to-lymphocyte Ratio (NLR), Platelet-to-lymphocyte Ratio (PLR) are more prominent in retinal artery occlusion (RAO) compared to retinal vein occlusion (RVO).PLOS ONE

Dear Dr. Lin,

Thank you for submitting your manuscript to PLOS ONE. After careful consideration, we feel that it has merit but does not fully meet PLOS ONE’s publication criteria as it currently stands. Therefore, we invite you to submit a revised version of the manuscript that addresses the points raised during the review process.

ACADEMIC EDITOR I participated as a reviewer for the initial evaluation of this manuscriptPlease improve your paper with the suggestions reported below

We look forward to receiving your revised manuscript.

Kind regards,

Antonio Pinna, M.D.

Academic Editor

PLOS ONE

Journal Requirements:

Academic Editor Comments:

The manuscript has been revised appropriately and with some minor suggested amendments noted below should be suitable for publication in PLOS ONE.

- Please incorporate in the Discussion your response to Reviewer #2 comment: "Although the pathophysiology of RAO and RVO are not the same, previous studies believed that they were both related to thrombosis and inflammation. Some studies suggest that arterial thrombosis contains more NETs than venous thrombosis. Although previous studies have shown NLR and/or PLR might be the potential biomarkers in RAO or RVO, we would like to explore if there is some difference between them."

- Line 125: please change "mean SD" into "mean ± standard deviation (SD)

- Line 126: please change "normalcy" into "normality"

- Line 127: please change "the postmortem test" into "post hoc analysis"

- Line 130: please change "to predict" into "predicted"

- Line 152: please move the full stop after (Figure 2)

- Line 200: please change "virous" into "various"

- Line 211: please change "Finally" into "To sum up"

- Line 215: please change "may be a potential biomarker for RAO rather than PLR" into "may be a more reliable biomarker for RAO than PLR"

- Throughout the test: please do not use the short form of auxiliary verbs. Use "we would" instead of "we'd", "did not find" instead of "didn't find", and so on.
---

## [Author Response · Author response to Decision Letter 1]

15 Jan 2022

RE: PONE-D-21-18118R1

Title: Neutrophil-to-lymphocyte Ratio (NLR), Platelet-to-lymphocyte Ratio (PLR) are more prominent in retinal artery occlusion (RAO) compared to retinal vein occlusion (RVO).

Academic Editor comments: 

1) Please incorporate in the Discussion your response to Reviewer #2 comment: "Although the pathophysiology of RAO and RVO are not the same, previous studies believed that they were both related to thrombosis and inflammation. Some studies suggest that arterial thrombosis contains more NETs than venous thrombosis. Although previous studies have shown NLR and/or PLR might be the potential biomarkers in RAO or RVO, we would like to explore if there is some difference between them." 

Response: We thank the editor for the comments. We have added that in the last paragraph. (page 13, lines 215-218).

2) - Line 125: please change "mean SD" into "mean ± standard deviation (SD) - Line 126: please change "normalcy" into "normality"

- Line 127: please change "the postmortem test" into "post hoc analysis"

- Line 130: please change "to predict" into "predicted" 

- Line 152: please move the full stop after (Figure 2)

- Line 200: please change "virous" into "various"

- Line 211: please change "Finally" into "To sum up"

- Line 215: please change "may be a potential biomarker for RAO rather than PLR" into "may be a more reliable biomarker for RAO than PLR" 

- Throughout the test: please do not use the short form of auxiliary verbs. Use "we would" instead of "we'd", "did not find" instead of "didn't find", and so on.

Response: We thank the editor for the comments. We have done all the revision above.

---

## [Editor Report · Decision Letter 2]

24 Jan 2022

Neutrophil-to-lymphocyte Ratio (NLR), Platelet-to-lymphocyte Ratio (PLR) are more prominent in retinal artery occlusion (RAO) compared to retinal vein occlusion (RVO).

PONE-D-21-18118R2

Dear Dr. Lin,

We’re pleased to inform you that your manuscript has been judged scientifically suitable for publication and will be formally accepted for publication once it meets all outstanding technical requirements.

Kind regards,

Antonio Pinna, M.D.

Guest Editor

PLOS ONE
---

## [Editor Report · Acceptance letter]

26 Jan 2022

PONE-D-21-18118R2 

Neutrophil-to-lymphocyte Ratio (NLR), Platelet-to-lymphocyte Ratio (PLR) are more prominent in retinal artery occlusion (RAO) compared to retinal vein occlusion (RVO)
Short title: NLR, PLR and MHR in Retinal Vascular Occlusions 

Dear Dr. Lin:

I'm pleased to inform you that your manuscript has been deemed suitable for publication in PLOS ONE. Congratulations! Your manuscript is now with our production department. 

Kind regards, 

on behalf of

Professor Antonio Pinna 

Guest Editor

PLOS ONE